# OpenReview forum: "Benchmarking Text-to-Image Safety: Using Adaptation Methods to Mitigate Oversexualization"
_TMLR — Rejected by TMLR_

### Review · Reviewer_qEeh · 2025-07-20

**Summary Of Contributions:**

1. This paper introduces the first systematic benchmark and evaluation framework specifically targeting the issue of oversexualization in text-to-image (T2I) generative models from benign prompts, whilre previous benchmarks all focus on safefy evaluation from the explict harmful prompts
2. To this end, the paper presents Benign2NSFW, a novel dataset of 1108 carefully curated prompts likely to elicit oversexualized outputs from models like Stable Diffusion v1.4.
3. The authors analyzed six adaptation strategies (2 inference-time methods: Reward Guidance and Safe Latent Diffusion; 4 fine-tuning methods: DRaFT, RWFT, PPO and DPO) with evaluation with human rater elo scores and automatic metrics like (CNN-based classifiers, CLIP-Score, etc). Their results reveal that although methods like PPO and DPO improve safety most effectively, they also introduce trade-offs in alignment and visual quality, emphasizing the need for multi-objective optimization in safe image generation.

**Audience:**

Yes

**Claims And Evidence:**

Yes

**Requested Changes:**

1. While Elo scores have proved their effectiveness in works like Chatbot Arena, it's not a direct reflection of how well a model perform on the benchmark absolutely. A simple idea of direct metrics of your Benign2NSFW can be the percentage that a T2I model generate NSFW images for the benchmark prompts, or the probabilitiy to generate NSFW image. Have you ever records these kinds of metrics during the experiments as it directly reflect how well a model perform, which the metrics in the paper (elo scores, CNN-based classififer scores) does not provide.
2. Please add some T2I model's performance on the benchmark like close source models (e.g. OpenAI DALLE-3, Google Veo3) and other open-source models like FLUX, Hyper-SD, Wanxiang, etc.
3. I have a question about the PPO part in the paper. The PPO differs itself with previous methods like TRPO by adding the clipped surrogate objective by cliping the advantage to a specific region. However, I don't see this part of definition in your training objective. Can you clarify this? Or in fact you are simply training on TRPO algorithm? [1]
4. Please try to bold some numbers in the table as highlights to make them more readable.

#### Reference

[1] Schulman, John, Filip Wolski, Prafulla Dhariwal, Alec Radford, and Oleg Klimov. "Proximal policy optimization algorithms." arXiv preprint arXiv:1707.06347 (2017).

**Strengths And Weaknesses:**

## Strengths
1. This paper is well-written and well-motivated. Figures and tables are nicely presented with corresponding analysis.
2. Instead of only proposing the benchmarks, the paper also explore potential ways to improve the model on the safety issues from benign prompts, where 2 inference-time methods and 4 fine-tuning methods are proposed. The analsis provide great insights for later models' development.
3. I appreciate the comprehensive experiments introduced, including multi-objective evaluation beside the safety, gender parity experiments, abaltion studies on the reward design and KL divergence in the appendix.

## Weaknesses
1. As a benchmark paper, it lacks evaluation of many existing T2I models and only focus on experiments based on SD v1.4, thus missing a potentail analysis of how well existing models (like DALL-E, Veo, Wanxiang, etc) perform on this benchmark. This is critical for the benchmark to be a general benchmark where later models can use for evaluation.
2. While 6 methods have been explored, it seems not a single method dominate on all the metrics, and most with a cost of decrease of alignment scores for higher safety scores. It's unclear how to make this trade-off according to the paper.
3. The metrics used in the benchmark, while comprehensive (e.g. elo scores, auto metrics like CNN classifier scores) does not directly reflect how many percentage of NSFW images a T2I models will generate on the benchmark prompts. People may find these metrics confusing and hard to be convinced.

---

### Review · Reviewer_QgUd · 2025-07-25

**Summary Of Contributions:**

This paper aims to address the issue of oversexualization in text-to-image diffusion models, where models generate more sexualized content than intended by benign user prompts.
It introduces the first comprehensive benchmark, Benign2NSFW, consisting of 1,108 neutral prompts designed to elicit oversexualization.
The study evaluates six adaptation methods (two inference-time methods and four fine-tuning methods) on Stable Diffusion v1.4.
Experiments show that:
1) PPO and DPO are most effective at reducing oversexualization, with PPO achieving the highest safety scores.
2) Trade-offs exist between safety and quality (e.g., PPO enhances safety but degrades aesthetics and artifacts).

**Audience:**

Yes

**Claims And Evidence:**

Yes

**Requested Changes:**

1. The paper uses LAION’s NSFW classifier and a CNN-based safety classifier to quantify "safety." This could be biased. What is the performance under different classifiers?
2. How were these classifiers validated for fairness across genders, cultures, or body types? For instance, do they disproportionately label images of women as "NSFW" compared to men, even for identical levels of exposure?
3. It is also encouraged to discuss different T2I models instead of just evaluating on the SD1.4 model.

**Strengths And Weaknesses:**

Strength
1. The introduction of the Benign2NSFW dataset fills a critical gap in existing safety benchmarks, which primarily focus on adversarial prompts.
2. The paper systematically compares six adaptation methods (inference-time and fine-tuning) across multiple metrics (safety, alignment, artifacts, aesthetics, and human preferences). This holistic analysis provides clear guidance for practitioners choosing between methods.

Weakness
1. All evaluations are conducted on the Stable Diffusion Model 1.4, which is already an out-of-fashion model. Whether the method can generalize to new models (FLUX, SDXL, etc.) and how these new models perform in this setting is unclear.
2. The safety evaluations rely heavily on open-source NSFW classifiers, which may have their own biases or limitations. This could introduce noise in measuring "safety" and affect the reliability of comparisons between methods.

---

> ### Comment · Action_Editor_S8Ks · 2025-09-02
>
> Dear reviewer,
>
> Can you input your final recommendation?
>
> Best,
> AE

---

### Review · Reviewer_eZcm · 2025-08-13

**Summary Of Contributions:**

The paper addresses the underexplored problem of oversexualization in generative text-to-image (T2I) diffusion models, where benign prompts unintentionally produce sexualized images. Such a problem could potentially be significant in practice and worth investigating. This paper benchmarks six adaptation methods, including two inference-time (Reward Guidance, Safe Latent Diffusion) and four fine-tuning approaches (DRaFT, PPO, DPO, RWFT), on their ability to reduce oversexualization, while also measuring impacts on safety, alignment, preference, artifacts, and aesthetics. Both automated classifiers and human evaluations are used to assess results. Through numerical evaluations, the performance of oversexualization has been sufficiently benchmarked and provided with several insights.

**Audience:**

Yes

**Broader Impact Concerns:**

No ethnical concerns.

**Claims And Evidence:**

Yes

**Requested Changes:**

Please see the weaknesses to justify my concerns.

**Strengths And Weaknesses:**

Strengths:
- The paper focuses on oversexualization as a distinct safety failure mode, which is well-motivated and not widely studied.
- Both inference-time and fine-tuning methods are provided with a balanced, representative view of available adaptation strategies.
- The authors examine the relationship between multiple dimensions (safety, quality, alignment, artifacts, preference), which is essential for understanding real-world applications.

Weaknesses:
- The paper studies a potentially interesting problem; however, there is no intuition on how to reduce oversexualization during methodology design and practical applications.
- How to quantify oversexualization is unclear. Such a problem requires a more formal analysis to provide a convincing protocol or criteria to quantify oversexualization.
- Using only 7 human raters is quite limited, which could lead to rating bias, thus raising concerns about the evaluation quality.

---

> ### Author Response · Authors · 2025-08-27
> **Thank you for your thorough and helpful feedback**
>
> Thank you for your thorough and helpful feedback. We address your comments below:
>
> **No new method to address oversexualization**: This paper introduces a novel prompt dataset to benchmark the oversexualization behavior of text-to-image models. The work does not propose a new adaptation method. Instead, it offers a crucial tool for evaluating models and informing the development of future adaptation techniques. The paper also provides a comparative analysis of existing adaptation methods by applying them to the new prompt dataset. This analysis yields insights into the strengths and weaknesses of these methods in mitigating oversexualization.
>
> **How to quantify oversexualization is unclear.**: Harm Amplification in Text-to-Image models proposes such a method (https://arxiv.org/pdf/2402.01787). Holistic Evaluation of Text-to-Image Models (https://arxiv.org/pdf/2311.04287) proposes the use of the NSFW classifier, and similar classifiers as measures of “toxicity”. The NSFW and racy classifiers used within this work follow that suit. Following the “Harm amplification” proposal, we can view “oversexualization” as occurring when an image is toxic, but a prompt is benign. In this way the paper follows existing work in measuring oversexualization as the toxicity in images, according to classifiers and human ratings around sexual content of images, for benign prompts.
> To quantitatively assess the benignity of our prompts, we employ an in-house model designed to measure the textual pornographic content. Our dataset's mean score is 0.1226, while the I2P dataset's mean is 0.1836. A two-sample t-test yielded a t-statistic of -9.1857 with a p-value of 1.05e-19, indicating a statistically significant difference between the two datasets.
>
> **Robustness of Human Evals**: We appreciate the reviewer’s concerns regarding the limited amount of human raters in our eval and its potential for bias. Due to time and resource constraints, we are unable to increase the number of raters, but we have conducted bootstrap analysis on the ELO scores to show the robustness of ratings. Using 1k bootstraps, we estimated the confidence intervals for each model’s ELO score for safety and assessed the frequency in which each model appears in the top 3 ranking. We present the results in the table below. Methods such as PPO, DPO, and SLD all occur frequently in the top 3 methods when conducting bootstrap analysis showing that their ELO score rankings are robust.
>
> | Method | ELO Score | 95% CI | Frequency in Top 3 |
> |---|---|---|---|
> | Baseline   |    977.67  | [962.59, 994.58]  | 0%  |
> | Reward Guidance    | 966.81    | [959.04, 993.61]   | 0%   |
> | SLD                | 1018.36    | [1001.69, 1035.01]    | 98% |
> | DRaFT              | 982.45     | [967.24, 995.96]     |  0%   |
> | PPO                | 1027.80 | [1009.06, 1047.75]    |  99%   |
> | DPO                | 1023.64    | [1005.93, 1040.61]     |  99%    |
> | RWFT               | 993.29    | [979.41, 1007.10] | 3%     |

---

### Review · Reviewer_FxSr · 2025-08-14

**Summary Of Contributions:**

This submission introduces a benchmark dataset called Benign2NSFW, consisting of 1,108 benign prompts designed to elicit oversexualized outputs from text-to-image (T2I) models like Stable Diffusion v1.4. It evaluates six existing adaptation methods (two inference-time: Reward Guidance and Safe Latent Diffusion; four fine-tuning: PPO, DPO, RWFT, and DRaFT) for mitigating oversexualization, using both human and automated evaluations across safety, alignment, aesthetics, and artifacts.



However, the contributions appear limited and incremental at best. The benchmark's construction is superficial, relying on vague red-teaming without empirical justification for why certain prompts trigger issues. No novel methods are proposed—all adaptations are pre-existing—and the observations (e.g., PPO improves safety but degrades quality) lack depth or surprising findings.

**Audience:**

Yes

**Claims And Evidence:**

Yes

**Requested Changes:**

To potentially salvage this for acceptance, major revisions are needed—these are mostly critical unless noted otherwise, as the current version lacks sufficient novelty and methodological soundness.



- Expand and Refine the Benchmark: Triple the dataset size to at least 3,000–5,000 prompts, drawing from larger sources. Introduce a detailed taxonomy, categorizing prompts by factors like object count, spatial relations, demographic attributes (e.g., intersectional biases beyond binary gender), and prompt complexity. Provide empirical evidence for prompt selection, such as ablations showing which linguistic features (e.g., adjectives like "curvy" vs. neutral descriptors) trigger oversexualization, with statistical analysis.
- Demonstrate Novelty in Methods or Insights: Propose at least one new adaptation variant (e.g., a hybrid of PPO and SLD with fairness-aware rewards) or conduct deeper analyses, such as causal interventions on why RG fails on benign prompts (e.g., using attribution methods like integrated gradients). Highlight non-obvious findings, perhaps via controlled experiments on prompt variations.
- Strengthen Evaluations and Fairness: Extend gender analysis to non-binary identities and intersections (e.g., race+gender). Use more diverse evaluators for human studies (beyond 7 raters) and validate automated metrics against larger baselines.



Without addressing the critical changes, particularly the benchmark's scale and structure, I cannot recommend acceptance—this submission does not meet the standards for innovative, rigorous work in T2I safety.

**Strengths And Weaknesses:**

**Strengths**

- The paper attempts a systematic comparison of six adaptation methods on a specific failure mode (oversexualization from benign prompts), which could serve as a starting point for practitioners interested in safety-quality trade-offs.
- Qualitative examples and practical lessons (e.g., KL regularization in PPO to prevent reward hacking) offer minor implementation insights.

**Weaknesses**

- Lack of Innovation: The core methods are all borrowed from prior works (e.g., PPO from Fan et al., 2023; DPO from Wallace et al., 2024), with no new algorithms, theoretical advancements, or meaningful modifications. The "contributions" boil down to assembling a benchmark and running evaluations, but without novel observations—e.g., no deep analysis of why certain methods fail on oversexualization beyond superficial trade-offs.
- Benchmark Quality and Scale: Benign2NSFW's construction is inadequately detailed and unconvincing. The "internal red-teaming" is vaguely described, with no protocols shared, no experimental evidence (e.g., ablation on prompt features causing oversexualization), and no justification for prompt selection beyond filtering with a text classifier. At only 1,108 prompts with an average length of 6.76 tokens, it is undersized and simplistic. Typical T2I prompt lengths in datasets like Midjourney (avg. dozens words) or DiffusionDB (dozens) far exceed this, making Benign2NSFW ill-suited for realistic image description scenarios.

- Absence of Taxonomy and Granularity: This benchmark lacks any structured taxonomy. Prompts are coarsely organized (e.g., top tokens like "person," "man," "woman" without sub-categories), failing to probe multifaceted aspects like object interactions, demographic intersections, or prompt complexity—resulting in a naive, low-resolution evaluation.
- Method Effectiveness and Insights: No method emerges as clearly superior or yields "interesting" observations; e.g., PPO boosts safety but introduces artifacts, a predictable trade-off without deeper causal analysis.  The paper overlooks broader implications, such as scalability to newer models like SDXL or Flux.1.

Overall Negative Assessment: The submission feels underdeveloped for a venue like TMLR, lacking the rigor, scale, and originality expected in a benchmark paper. It does not convincingly advance knowledge on T2I safety beyond aggregating known techniques.

---

> ### Author Response · Authors · 2025-08-29
> **Thank you for your insights and suggestions.**
>
> We address your comments below:
>
> **Expand and Refine the Benchmark**:
> In response to the reviewer's feedback, we conducted a more granular analysis of our prompt dataset. We've enhanced our annotation scheme by categorizing prompts across four key attributes:
>
> 1. Race/Ethnicity: Not Specified, White, Black, Asian, South Asian, Latino, Other, Non-human
> 2. Gender: Not Specified, Male, Female, Transgender: man, Transgender: woman, Transgender: unspecified, Other
> 3. Sexual Orientation: Not Specified, Straight, Gay, Lesbian, Other
> 4. Skin Tone: Not Specified, Light, Medium, Dark
>
> | Race  | Count |
> |-------------|-------|
> | Not Specified   | 947  |
> | White | 30  |
> | Black | 43 |
> | Asian  | 29  |
> | South Asian  | 8  |
> | Latino | 19  |
> | Other   | 2  |
> | Non-human  | 30  |
>
> | Gender | Count |
> |-------------|-------|
> | Not Specified | 507 |
> | Male | 277 |
> | Female | 256 |
> | Transgender: man | 19 |
> | Transgender: woman | 26 |
> | Transgender: unspecified | 11 |
> | Other | 12 |
>
> | Sexual Orientation  | Count |
> |-------------|-------|
> | Not Specified  | 969  |
> | Straight | 22 |
> | Gay  | 62 |
> | Lesbian | 54  |
> | Other  | 1  |
>
> | Skin Tone   | Count |
> |-------------|-------|
> | Not Specified  | 1079 ||
> | Light  | 3 |
> | Medium | 1  |
> | Dark  | 25  |
>
> **Expanded Fairness Analysis**
> We have expanded our fairness analysis to include intersections of race (Asian, White, Black) and gender (Female, Male).  Notably, most methods except PPO have a lower safety score on images generated for Asian female prompts.
>
> | Race | Gender | Baseline | RG | SLD | DRaFT | PPO | DPO | RWFT |
> |---|---|---|---|---|---|---|---|---|
> | Asian | Female | 0.471 | 0.458 | 0.593 | 0.489 | 0.953  | 0.491  | 0.500  |
> | Asian | Male | 0.609 | 0.610 | 0.736  | 0.649  | 0.895  | 0.616  | 0.596  |
> | Black | Female | 0.715 | 0.717 |  0.777 | 0.733  | 0.956  | 0.704  | 0.743  |
> | Black | Male | 0.707 | 0.704 | 0.808  | 0.716  | 0.900  | 0.721  | 0.706  |
> | White | Female | 0.584 | 0.618 | 0.731  | 0.694  | 0.873  |0.651   | 0.681  |
> | White | Male | 0.828 | 0.824 | 0.863  | 0.808  | 0.966  |0.786   |0.826   |
>
> **Benchmark Quality and Scale**:
> The I2P (https://huggingface.co/datasets/AIML-TUDA/i2p) dataset, used as a benchmark dataset for measuring T2I toxicity across areas of Hate/Harassment/Violence/Self-harm/Sexual content/Shocking image/Illegal activity, as in the benchmarking paper Holistic Evaluation of Text-to-Image Models (https://arxiv.org/pdf/2311.04287), consists of 4703 prompts of which only 765 were sexually explicit. Our benchmark dataset of 1108 prompts is comparable in size given its much narrower focus on specifically oversexualization - benign prompts yielding sexual imagery. Additionally, due to the filtering done to ensure the prompts are benign and images sexual, described in the paper, we see a bias towards shorter prompts, as underspecified prompts are more likely to elicit an image that is disproportionately sexual to the prompt.
>
> **Demonstrate Novelty in Methods or Insights**: In this paper, we introduce a novel prompt dataset designed to systematically evaluate the oversexualization tendencies of text-to-image (T2I) models. To our knowledge, this is the first publicly available dataset of its kind. The primary contribution of this work is the dataset itself, which serves as a crucial tool for assessing model safety and bias. The second contribution is to perform a comparative analysis of existing adaptation methods using this new benchmark and evaluate their effectiveness in mitigating oversexualization. We emphasize that the focus of this paper is on the comparative evaluation of these methods, not the proposal of a new adaptation technique.
>
> Our findings indicate that no single adaptation method consistently outperforms others across all evaluation metrics when finetuned and tested with our novel oversexualization prompt dataset. This highlights a critical need for further research into developing more robust and effective adaptation techniques, which is out of scope of this study. The necessity of our benchmark dataset for evaluating these new methods is also underscored by these results.
>
> For our baseline model, we intentionally selected Stable Diffusion v1.4. This choice was strategic, as SD v1.4 is known to be particularly prone to generating sexually explicit images, making it an ideal candidate for demonstrating the impact of our work. By applying various adaptation methods in conjunction with our dataset, we can clearly showcase the potential for significant improvement in model safety, even starting from a highly vulnerable baseline.
> Below are automated metrics on Benign2NSFW for SD v1.4, SDXL, and Flux.1. SD v1.4 tends to generate more unsafe images (*lower* safety scores and *higher* NSFW scores mean *more* sexualized) providing a good foundation for our study.
>
> | Model | Safety Score | NSFW Score |
> |---|---|---|
> | SD v1.4 | 0.774 | 0.233 |
> | SDXL    | 0.805 | 0.106 |
> | Flux.1  | 0.866 | 0.161 |

---

### Decision · Action_Editor_S8Ks · 2025-09-21

**Recommendation:** Reject

**Audience:**

Yes

**Audience Explanation:**

Characterization of T2I models in certain aspects is interesting to researchers on image generative models.

**Claims And Evidence:**

No

**Claims Explanation:**

This paper aims to address the issue of oversexualization in text-to-image diffusion models, where models generate more sexualized content than intended by benign user prompts. It introduces the first comprehensive benchmark,Benign2NSFW, consisting of 1,108 neutral prompts designed to elicit oversexualization. The study evaluates six adaptation methods (two inference-time methods and four fine-tuning methods) on Stable Diffusion v1.4.

After the rebuttal, three reviewers gave “Leaning to Reject” and one reviewer gave “Leaning to Accpet”. All reviewers acknowledge the new problem setting of measuring oversexualization of T2I images. The criticisms focus on only SD 1.4 being used (solved by adding SDXL and FLUX.1 in the rebuttal), small-scale prompts (only ~1100 prompts), using existing classifiers to qualify oversexualization not being fully justified, a small number of human raters. Most concerns are not solved after the rebuttal. Therefore, the AE recommends rejection of this paper at its current format.

The AE ignores the comment of lacking novelty on methodology, as TMLR reviews shall focus on validating claims.

**Resubmission Of Major Revision:**

The authors may consider submitting a major revision at a later time.